# LMM4-IC4K: A Large Multimodal Model Powered Integrated Circuit Footprint Geometry Understanding

Yida Wang [* 1]  Taiting Lu [* 2]  Runze Liu [2]  Lanqing Yang [1]  Zhe Chen [1]  Yuehai Wang [1]  Yixin Liu [1]
Kaiyuan Lin [2]  Xiaomeng Chen [3]  Dian Ding [1]  Yijie Li [4]  Yifan Yang [5]  Yi-Chao Chen [1]  Yincheng Jin [6]
Mahanth Gowda [2]

## Abstract

Printed-Circuit-board (PCB) footprint geometry labeling of integrated circuits (IC) is essential in defining the physical interface between components and the PCB layout, requiring precise visual perception. However, the unstructured nature of footprint drawings and abstract diagram annotations prevents direct IC footprint parsing and automated package geometry labeling methods from developing. Existing Large Multimodal Models (LMMs) struggle with inaccurate geometric perception, limiting their effectiveness in this task. To address these challenges, we propose LMM4-IC4K, a novel framework that treats IC mechanical drawings as images and leverages LMMs for structured geometric interpretation. To support such a framework, we introduce ICGeo8K, a multi-modal dataset with 8,608 labeled samples, including 4138 real-world IC footprint samples and 4470 synthetically generated samples. We further present a two-stage training framework to fine-tune LMMs for IC footprint labeling. Extensive experiments demonstrate that our model outperforms state-of-the-art LMMs on the proposed benchmark. The accurate translation of footprint diagrams enabled by LMM4-IC4K contributes to advancing automation and standardization within the PCB industry.

---

[*]Equal contribution  [1]Computer Science & Technology, Shanghai Jiaotong University, Shanghai, China [2]Computer Science & Engineering, Pennsylvania State University, Pennsylvania, USA [3]Department of Electronic Engineering, Shanghai Jiaotong University, Shanghai, China [4]School of Computing, National University of Singapore, Singapore [5]Microsoft, Shanghai, China [6]Computer science, State University of New York at Binghamton, New York, USA. Correspondence to: Lanqing Yang <yanglanqing@sjtu.edu.cn>, Yifan Yang <yifanyang@microsoft.com>.

*Proceedings of the $43^{rd}$ International Conference on Machine Learning*, Seoul, South Korea. PMLR 306, 2026. Copyright 2026 by the author(s).

## 1. Introduction

IC footprint geometry understanding has significant value in industrial production and PCB design because it ensures accurate component placement and reliable electrical connections on the PCB. In industrial practice, IC footprint pins are illustrated in human-readable diagrams and define where and how the IC connects to the PCB board, determining both the physical placement and the electrical pathways needed for the circuit to function properly (as depicted in Figure 1). Inaccuracies in pin size or placement can lead to mismatched impedance or unintended parasitic effects. These issues can significantly degrade the performance and reliability of high-speed or sensitive circuits (Circuits, 2023; Corporation, 2023). There have been numerous industrial electronic design automation (EDA) tools, such as IPC compliant footprint wizard by Altium (Altium Limited, 2022), Package Generator by Autodesk EAGLE (Autodesk, Inc., 2020), and Footprint Editor by KiCad (SparkFun Electronics, n.d.), etc. While existing tools support manual footprint creation, they typically demand substantial user input and offer limited automation in interpreting datasheet diagrams. Furthermore, engineers often rely on personal experience to construct EDA footprints, frequently deviating from the annotations specified in IC datasheets. This practice contributes to non-standardized footprint production and increases the risk of imperfect electrical connections, ultimately leading to costly rework and reduced design reliability (ASSEMBLY, 2021).

Understanding a precise IC footprint diagram (example shown in Figure 2) involves three core tasks: (1) Identifying **the number of pins** correctly to ensure accurate electrical connectivity. (2) Accurately determining **the spatial arrangement of all pins** relative to a clearly defined reference point (origin), such as the component's center or a specific pin, to ensure correct placement and alignment of the component on the PCB. (3) Determining **the exact size of each pin** is important to make sure the electronic parts are firmly attached and work properly. These properties require complex relational reasoning between abstract diagrammatic symbols and numerical annotations, a task that simple detection algorithms or optical character recognition

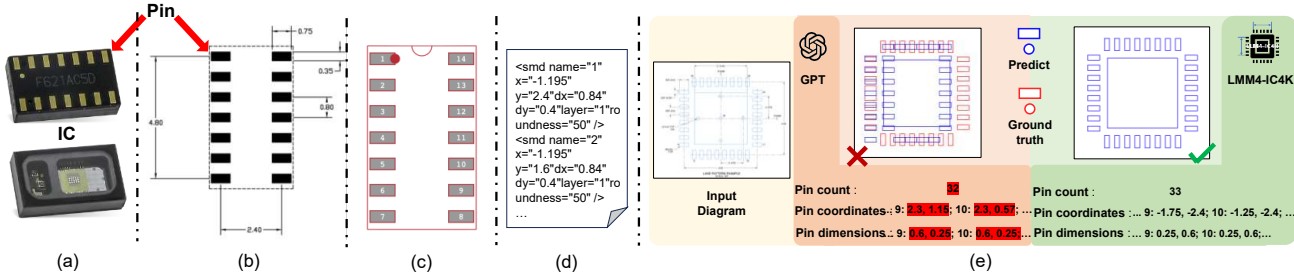

*Figure 1.* Illustration of IC geometry understanding problem. (a) Example IC packages with visible pins, serving as electrical and mechanical interfaces. (b) Corresponding datasheet footprint diagram showing pin positions and dimensions for PCB design. (c) Visualization of EDA footprint based on the datasheet diagram. (d) Raw EDA footprint source file. (e) Visualization of an IC footprint parsing task: GPT and LMM4-IC4K are given the same input diagram and tasked with extracting pin count, pin center coordinates, and pin dimensions. GPT produces incorrect answers, with errors highlighted in red, such as miscounted pin numbers or incorrect coordinates.

(OCR) methods are ill-equipped to handle effectively.

LMMs have shown promising performance in geometry and spatial reasoning (Gao et al., 2023a; Cheng et al., 2024), proof and logic reasoning (Zheng et al., 2023), and Graph and Set Theory (Wang et al., 2024a). However, existing research mainly focuses on abstract geometric reasoning within textual and synthetic visual domains, whereas complex, real-world engineering tasks, such as understanding PCB footprint geometry and IC package drawings, remain underexplored. Since general LMMs are not designed to handle dense IC footprint diagrams that include many annotations and engineering drawing labels (shown in Figure 2), they struggle to perform accurately on such visually complex tasks. Achieving a precise understanding of the IC footprint geometry labeling problem with LMMs necessitates high-quality instructional prompts and accurately aligned image-label data pairs, which in turn **require meticulous, pin-level annotations**, Constructing such trainable diagram-description pairs demands substantial manual effort and domain expertise. Moreover, no publicly available datasets currently exist that provide IC diagrams suitable for directly training LMMs in IC geometry understanding.

In this paper, we propose LMM4-IC4K (**L**arge **M**ultimodal **M**odel for (**4**) **I**ntegrated **C**ircuit - 4K), an LMM-empowered framework designed to accurately understand IC footprint geometry and automate the footprint generation process by mimicking the step-by-step reasoning approach used by human engineers. We construct a reasoning dataset comprising real-world footprint diagrams extracted from datasheets, as well as synthetically generated footprint drawings, to effectively fine-tune an LMM for expertise in IC footprint geometry understanding.

To rigorously evaluate the performance of existing LMMs and our approach in IC footprint labeling, we construct a systematic benchmark that not only captures real-world complexities, such as varying pin counts and package styles, but also aligns closely with the actual distribution of IC footprint types observed in real-world datasets, covering all

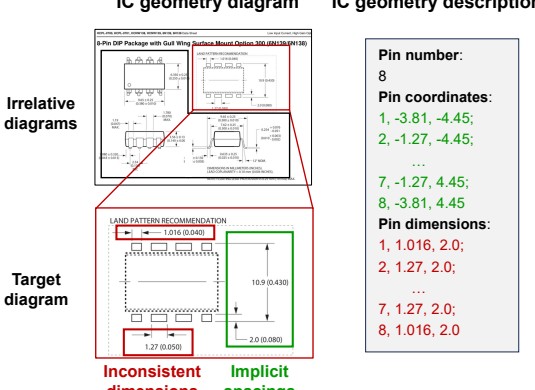

*Figure 2.* An example of IC geometry understanding. This starts with locating the correct diagram (upper-right corner). Then the dimensions (red texts) and locations (green texts) of pins are extracted and computed. Datasheet pages often contain irrelevant and misleading information (*e.g.*, the 3-view diagrams, inconsistent units, and implicit spacing labels), adding to the complexity.

major package categories (detailed in Section 3). The main contributions of this article are threefold:

- To our best knowledge, **we are the first to develop a novel multimodal geometric reasoning dataset for IC footprint labeling, ICGEO8K**, containing a total of 8,608 labeled samples–4,138 collected from real-world IC footprint diagrams and 4,470 synthetically generated. The dataset contains three sub-tasks, each targeting a key aspect of geometric reasoning: (i) counting the number of pins, (ii) computing the center coordinates of individual pins, and (iii) estimating the dimensions of each pin.

- **We introduce a novel benchmark on IC footprint geometry understanding, ICGEOQA**, and evaluate the performance of state-of-the-art general-purpose LMMs on this task. These models demonstrate limited capabilities in precise geometric reasoning, resulting in low layout accuracy and significant errors in pin localization and dimension estimation. Furthermore, we

examine the discrepancies between manually created EDA descriptions and corresponding datasheet diagram annotations, introducing a manual baseline for IC footprint labeling.

- **We propose LMM4-IC4K, an LMM-empowered framework designed to accurately understand IC footprint geometry and automate the footprint generation process**. Our approach introduces a two-stage training paradigm that enables LMMs to interpret IC footprint diagrams and generate corresponding designs effectively. LMM4-IC4K achieves state-of-the-art performance on a real-world benchmark, outperforming existing LMM-based methods and surpassing the industrial manual baseline by 62.7%. Additionally, it achieves a 28× speedup in labeling time and a 435× cost efficiency gain. The results demonstrate LMM4-IC4K's precision, efficiency, and promise in standardizing IC footprint labeling for the PCB industry.

Codes and datasets are available in https://github.com/IC-LMM/LMM4-IC4K.

## 2. Related Work

**Manual PCB Component Footprint Generation.** Traditional PCB component geometry generation is labor-intensive and time-consuming, requiring manual interpretation of datasheets, footprint creation, symbol generation, and signal mapping (Ni et al., 2020; Engineer, 2021). With hundreds of components in modern designs, this manual process becomes a bottleneck (Martens, 2022), prone to inconsistencies, human error, and outdated libraries due to frequent spec updates (Teel, 2023; Abraham, 2025). The iterative nature of PCB design further complicates manual updates, delaying time-to-market. These limitations call for automated solutions to improve efficiency and reliability.

**Automated IC Footprint Geometry Understanding.** Existing PCB labeling methods focus on the segmentation or classification of IC footprints (Ni et al., 2020; Yang et al., 2024). These methods do not attempt to understand the geometric information of IC pins, leaving the automated labeling process limited to a level above the individual pins. Although object detection methods (Jocher, 2020; Tan et al., 2020; Zhu et al., 2020) may help with counting the number and computing the relative size of the IC pins, they cannot handle footprint diagrams with implicit information omitted, as shown in Figure 1(d). OCR methods (Du et al., 2024; Duan et al., 2025) can be used to extract diagram texts from datasheet images, but they cannot understand the physical and geometrical meaning of numerical labels. Some works utilize object detection and OCR for diagram object detection (Hu et al., 2023) or converting legacy schematic diagrams (Nurminen et al., 2020). However, these methods

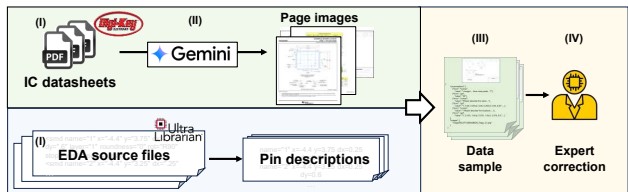

*Figure 3.* Dataset building process. Valid samples in ICGEO8K are developed in four steps: (I) datasheets and EDA files collection, (II) AI-aided diagram page finding, (III) image and label processing, and (IV) expert label corrections.

fail to bridge the gap between the extracted annotations and the implicit geometric knowledge. Therefore, a method capable of performing logical reasoning by interpreting both the footprint diagrams and annotation information is needed for fully automated IC footprint geometry labeling.

**LMM for Mathematics.** General-purpose LMMs such as ChatGPT, DeepSeek, and Qwen2 have demonstrated strong generalization abilities across various tasks without the need for task-specific fine-tuning (Bubeck et al., 2023; Bai et al., 2023; Guo et al., 2025). To address the problem of visual geometry reasoning, vision-language models like LLaVA, UniChart, and MathVista use large-scale image-text datasets to develop broad visual reasoning capabilities (Lu et al., 2023; Masry et al., 2023; Gao et al., 2023b; Lu et al., 2026). However, no existing work has explored the capabilities of LMMs in understanding IC footprint geometry.

## 3. Data Collection

Building an IC footprint dataset is challenging due to the lack of a complete index of IC models and the absence of a unified source that provides both datasheets and corresponding EDA footprint files. This fragmentation requires collecting and aligning data from multiple platforms, making large-scale data acquisition labor-intensive, error-prone, and difficult to automate. To build a comprehensive IC footprint dataset, we follow a four-step pipeline (shown in Figure 3) by integrating resources from multiple platforms, as no single source provides all the necessary data.

### 3.1. Dataset Building

**Phase I: Datasheet and EDA collecting.** Our dataset consists of paired samples, comprising IC footprint diagrams as input data and corresponding IC geometry descriptions as structured labels. An example of IC footprint geometry labeling is shown in Figure 2. To construct such dataset, we first extract a broad list of IC models from Digi-Key (Electronics, 2025), a global distributor offering over 13 million electronic products from more than 2,000 manufacturers. IC part numbers are used as unique identifiers to ensure consistent alignment across different data sources. We then collect the corresponding datasheets from Digi-Key, which

provide essential design information such as footprint diagrams, mechanical package types, and reference application circuits. Due to the limited availability of EDA resources on Digi-Key, we augment the dataset by retrieving standardized footprint source files, based on the part number from Ultra-Librarian (Ultra Librarian, 2024), a widely used platform for manufacturer-approved EDA content. The IC geometry descriptions, including properties such as pin count, center coordinates, and physical dimensions, are extracted from the EDA source files and aligned with the footprint diagrams.

**Phase II: AI-aided diagram page finding.** Because datasheets use different formats, terms, and visual styles, and often include multiple footprint diagrams for different versions of the same chip, it is difficult to directly use them as input for training without first carefully processing and selecting the correct IC footprint images. To address this challenge, we utilize Gemini 2.0 (Mallick & Kilpatrick, 2025) to assist with identifying the datasheet pages that contain IC footprint diagrams. Given that the model's predictions are not always precise, manual verification and correction by expert engineers are conducted to ensure accurate localization. The verified pages containing suggested pad layout diagrams are then extracted and used as image inputs for downstream IC geometry understanding tasks.

**Phase III: Image and label processing.** After identifying and verifying the page number containing the IC footprint diagrams in the datasheet, we extract those pages as images to serve as the input data. In parallel, we preprocess the corresponding EDA source files to extract structured pin-level information, as they often include extra, unrelated design details such as general component information, wiring data, and board layout elements. To focus only on the footprint geometry, we extract and organize three key properties: (i) the total pin count, (ii) the center coordinates of each pin, and (iii) the size of each pin. These are then formatted to match the label structure shown in Figure 2.

**Phase IV: Expert data correction.** Although the number of pins and their relative positions in the EDA file are generally consistent with the IC footprint diagrams in the datasheet, discrepancies often arise in the exact pin dimensions and precise locations. These inconsistencies, caused by variations in EDA file formatting and design conventions, can prevent LMMs from accurately understanding or inferring the geometric details found in EDA files solely from the visual information in datasheet diagrams. Therefore, we engage human engineering experts to review and correct any inaccurate labels in the EDA files, ensuring that they precisely match the specifications outlined in the IC geometry diagrams. All PCB engineers involved are paid $0.61$ per processed sample. After manual correction and filtering, we carefully selected $4,138$ IC footprint entries, spanning 10 distinct IC package types, each consisting of

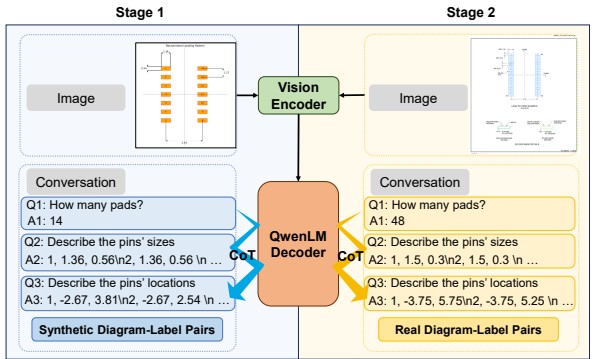

*Figure 4.* Two-stage training overview. We apply a two-stage supervised fine-tuning on Qwen2-VL-7B. The QAs in a conversation are ordered in a CoT manner.

datasheet-oriented IC footprint diagrams and their corresponding geometry description labels.

### 3.2. Synthetic Diagram Augmentation

Due to the time- and labor-intensive nature of manual labeling, collecting a large-scale real-world dataset is challenging, which in turn limits the effectiveness of model fine-tuning. To mitigate data scarcity, we developed a footprint diagram generation toolkit (detailed in Appendix A.5) that synthesizes clean, datasheet-style footprint images from real EDA geometry descriptions collected from Ultra Librarian CAD models. This approach enables the construction of a synthetic dataset, wherein the synthetic images serve as inputs and the corresponding EDA footprint descriptions provide accurate ground truth annotations for training. The synthetic dataset contains 4,470 data samples synthesized from EDA files collected from **Phase I** and adheres to the real-world IC distribution. Combined with $4,138$ real-world IC footprint diagram-label pairs, our final dataset ICGEO8K comprises a total of 8,608 entries.

## 4. Training Pipeline

To realize the geometric reasoning ability for understanding IC footprint diagrams, this section introduces our proposed LMM4-IC4K in detail. The main intuition is to leverage the capabilities of LMMs to perceive the relationships between image patterns and annotations, and to derive effective information progressively, similar to how humans process and reason through such information. To achieve this goal, we propose a two-stage, end-to-end IC geometry labeling framework (Figure 4) that progressively extracts pin geometries through a chain-of-thought (CoT) manner.

### 4.1. Supervised Fine-tuning with Chain-of-thought

When human engineers draw EDA footprints based on the IC footprint diagrams, several critical pieces of information need to be extracted from the diagrams, including the type

of IC, the total number of pins, the locations of the pins, and the dimensions of the pins. The thinking process in this procedure follows a progressive approach from the whole to the details: first, the engineer identifies the suggested pin layout diagram in the datasheet and determines the type of IC. Next, the engineer counts the number of pins and identifies their indices. Then, the engineer identifies and reads the spacings between the pins for positioning, followed by determining the pins' widths and lengths to define their dimensions. Inspired by this reasoning approach, LMM4-IC4K should also mimic this thinking process to derive the geometric information in a step-by-step manner.

To imitate human reasoning logic, we structure each query conversation for an IC geometry labeling task in three progressive questions: (1) *the number of pins in the IC footprint diagram*, (2) *the coordinates of each pin relative to the center of the diagram*, and (3) *the dimensions of each pin in millimeters*. **Question 1** directs the LMM to locate the correct diagram and identify the pins within the diagram. **Question 2** guides the LMM to identify each pin in index order and learn how to locate and interpret the corresponding annotations that reflect pin spacings. **Question 3** guides the LMM to learn how to identify and interpret the corresponding annotations that reflect pin dimensions. Example query conversations are shown in Figure 4.

### 4.2. Two-stage Training

To fully fine-tune the model for IC geometry understanding, we propose a two-stage, end-to-end training process. The LMM is initially trained on the synthetic samples of ICGEO8K. Since the IC diagrams in the synthetic part are generated from EDA descriptions using a plotting toolkit, the annotation rules and graphic patterns are relatively simple and uniform. This allows the LMM to acquire basic geometric reasoning capabilities without interference from complex real-world distractions. After the first round of training, the LMM undergoes further training on the real-world samples of ICGEO8K, where the target diagrams are embedded within datasheet pages. These real-world diagrams vary in image resolution, annotation rules, label text fonts, and presentation styles (detailed in Appendix A). Additionally, multiple diagrams may appear on the same datasheet page, increasing the complexity of identifying the desired diagram. During the second training stage, the model adapts to complex real-world scenarios, enhancing its ability to understand IC footprint geometric information.

## 5. Experiments

### 5.1. Dataset Analysis

We utilize the proposed ICGEO8K dataset to train the LMM. The package type of ICs can be categorized into 10 cate-

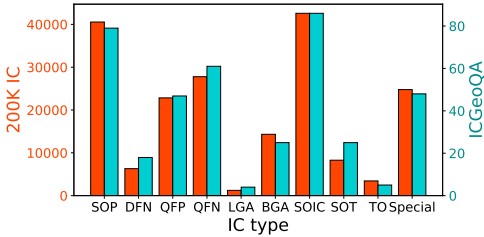

*Figure 5.* Package type distribution of ICGEOQA and 200K Digi-Key entries.

gories based on their pin patterns, based on established standards and guidelines provided in (IPC, 2010; Texas Instruments, 2023). Our dataset covers all package categories, reflecting the real-world distribution of common IC footprints. ICGEO8K possesses IC footprints with pin counts ranging from 1 to 800, implying the complexity in scales for IC pin labeling.

As no prior work addresses pin-level IC geometry labeling, we introduce a novel benchmark, **ICGEOQA**, to systematically evaluate the performance of LMMs in understanding IC footprint geometry. ICGEOQA consists of 400 carefully curated real-world IC entries sampled from ICGEO8K. To ensure that the benchmark reliably reflects real-world IC diversity, we analyzed the package type distribution across 200K IC entries crawled from Digi-Key. As shown in Figure 5, ICGEOQA exhibits an identical package category distribution to the full 200K entries and shares the same pin count distribution. These validations demonstrate that ICGEOQA reflects the distribution of real-world IC geometries and satisfies the needs of PCB engineers. Note that the **IC entries in ICGEOQA are excluded from ICGEO8K during the training process to avoid data leakage**. Please refer to Section A for further detailed descriptions of our benchmark and datasets.

### 5.2. Experiment Setup

**Implementation Details.** We fine-tune a footprint geometry understanding model based on Qwen2-VL (Wang et al., 2024b), a SOTA LMM in image understanding with Naive Dynamic Resolution mapping and Multimodal Rotary Position Embedding (M-ROPE). As a balance between performance and computational cost, we choose its 7B version (Qwen2-VL-7B) as our base model. We utilize LLaMA-Factory (Zheng et al., 2024) to fine-tune our model. As stated in Section 4.2, we first implement SFT on the LMM with the synthetic training samples for 3 epochs, and then fine-tune the LMM with real-world samples for 3 epochs. We use Low-Rank Adaptation (LoRA) for model training and set the cut-off length to 4096 and the learning rate to $5e^{-5}$. During each training stage, we randomly split 10% training samples as a validation set. All experiments are conducted on 2 NVIDIA A100-40G GPUs and repeated

*Table 1.* Comparison of QA performance with general LMMs on 3 tasks. Manual EDA serves as the industry-standard baseline for evaluating functional labeling accuracy by comparing real-world EDA descriptions against corresponding footprint diagram annotations in IC datasheets. "CV+OCR+Rule-base" method is a reference baseline for traditional CV approaches. The best-performing LMM is **in-bold**, and the second-best is underlined.

| Methods | Overall ($IoU_{IC}$ %) ↑ | Task 1 | | Task 2 ($d_{pin}$) ↓ | Task 3 ($IoU_{pin}$ %) ↑ |
| --- | --- | --- | --- | --- | --- |
| | | MAE ↓ | RMSE ↓ | | |
| **1-shot General LMMs** | | | | | |
| GPT-4o (Hurst et al., 2024) | $11.1 \pm 0.4$ | $8.21 \pm 0.47$ | $23.04 \pm 0.22$ | $4.01 \pm 0.02$ | $45.6 \pm 0.3$ |
| GPT-5 (Singh et al., 2025) | $40.6 \pm 1.1$ | $0.39 \pm 0.04$ | $4.13 \pm 0.19$ | $4.18 \pm 0.22$ | $65.4 \pm 0.1$ |
| Gemini 2.0 (Mallick & Kilpatrick, 2025) | $4.5 \pm 0.1$ | $1.84 \pm 0.40$ | $7.87 \pm 0.85$ | $18.27 \pm 0.45$ | $57.3 \pm 0.4$ |
| Gemini 2.5 Flash (Comanici et al., 2025) | $16.5 \pm 0.3$ | $11.13 \pm 0.70$ | $35.91 \pm 0.21$ | $3.56 \pm 0.15$ | $35.8 \pm 0.6$ |
| DeepSeek-VL2 (Wu et al., 2024) | $1.5 \pm 0.8$ | $21.97 \pm 0.35$ | $41.70 \pm 2.15$ | $4.22 \pm 0.23$ | $20.1 \pm 0.2$ |
| Qwen2-VL-7B (Wang et al., 2024b) | $1.7 \pm 0.1$ | $19.17 \pm 0.37$ | $43.76 \pm 0.84$ | $3.63 \pm 0.54$ | $41.1 \pm 0.5$ |
| **4-shot General LMMs** | | | | | |
| GPT-5 | $43.0 \pm 0.2$ | $\underline{0.37 \pm 0.05}$ | $\underline{3.11 \pm 0.19}$ | $4.12 \pm 0.31$ | $\underline{67.8 \pm 0.7}$ |
| Gemini 2.5 Flash | $38.0 \pm 0.7$ | $2.11 \pm 0.44$ | $8.72 \pm 1.07$ | $3.29 \pm 0.38$ | $57.2 \pm 0.2$ |
| Qwen2-VL-7B | $3.8 \pm 0.3$ | $12.99 \pm 0.20$ | $38.58 \pm 0.18$ | $3.51 \pm 0.31$ | $46.8 \pm 0.2$ |
| **Traditional Baselines** | | | | | |
| Manual EDA | $\underline{44}$ | - | - | $\underline{2.98}$ | $58$ |
| CV + OCR + Rule-based | $2.1$ | $19.28$ | $33.88$ | $57.64$ | $0.1$ |
| **Expert-finetuned Method** | | | | | |
| **LMM4-IC4K (ours)** | **71.6±0.5** | **0.35±0.07** | **2.81±0.08** | **1.11±0.02** | **88.0±0.3** |

three times (mean/std reported). The batch size is set to 2 per GPU at both training stages. Please refer to Appendix C.6 for detailed computational analysis.

**Evaluation Metric.** As no existing work focuses on pin-level IC geometry labeling, we develop our own metric, $IoU_{IC}$, defined as:

$$IoU_{IC} = \frac{Area_{pred} \cap Area_{label}}{Area_{pred} \cup Area_{label}}, \quad (1)$$

where $Area_{pred}$ is the area of the pin layout reconstructed from the predicted pin geometry, and $Area_{label}$ is the area of the pin layout reconstructed from ground truth pin geometry. $IoU_{IC}$ ranges from 0 to 1, where a value of 1 indicates the predicted pin layout completely overlaps with the ground truth pin layout, meaning the LMMs provides the correct answer precisely. A value of 0 indicates no overlap between predicted and ground truth layouts, denoting a completely incorrect result.

As stated in Section 4, the IC pin geometry is described by three sub-questions: pin counts, pin positions, and pin dimensions. Therefore, we also evaluate model performance for each sub-task. For **task 1 (pin counting)**, we use Mean Absolute Error (MAE) and Root Mean Square Error (RMSE), where lower values indicate a more accurate counting outcome. For **task 2 (pin positions)**, we calculate the Euclidean distance between each predicted pin's center coordinates and its corresponding ground truth. Then, the distances of all pins are averaged as $d_{pin}$, where a lower value indicates a more accurate position prediction. For **task 3 (pin dimensions)**, we use the average IoU between the predicted and ground truth pattern of each pin (center aligned), $IoU_{pin}$, as the evaluation metric. A more accurate dimension prediction achieves an $IoU_{pin}$ closer to 1. All metrics are calculated in mm unit.

### 5.3. Comparison with General LMMs

To evaluate the IC geometric reasoning capability of LMM4-IC4K, we make comprehensive comparisons between current SOTA general LMMs with LMM4-IC4K on ICGEOQA. All general LMMs are called via API. To ensure the outputs of general LMMs are compatible and have a consistent format as that of LMM4-IC4K, we utilize **single-shot prompt engineering** to inform the LMMs of the output formats.

We first compare LMM4-IC4K against a manual baseline evaluated from industrial libraries (detailed in Appendix C.2). As professional engineers occasionally deviate from exact datasheet specifications (see Section 3.1), this baseline provides a realistic benchmark for aligning footprint description mismatches with industrial requirements. LMM4-IC4K achieves an $IoU_{IC}$ of 71.6%, outperforming the manual baseline by 62.7%. This result highlights LMM4-IC4K's effectiveness in accurately labeling IC geometries and its strong capability to faithfully reconstruct the geometries described in footprint diagrams.

All general-purpose LMMs yield $IoU_{IC}$ scores below the manual baseline, indicating a failure to effectively perform IC geometry labeling. Since general LMMs do not have prior knowledge of IC footprint geometry labeling, they fail to understand the meaning of the numerical annotations and graphic symbols in the diagrams, such as mistaking pin spacing annotations as dimensions and miscounting pin numbers with omission symbols. Moreover, general LMMs struggle in accurately identifying numbers, resulting in great misinterpretations of geometric information. In contrast, LMM4-IC4K acquires prior knowledge on IC footprint diagrams, gaining the capability of labeling IC footprint geometry accurately. The improvement in $IoU_{IC}$

*Table 2.* Comparison of QA performance with different training strategies and dataset settings. The best-performing model in each experiment is **in-bold**, and the second-best is underlined.

| Factor | Strategy | Overall ($IoU_{IC}$ %) ↑ | Task 1 | | Task 2 ($d_{pin}$) ↓ | Task 3 ($IoU_{pin}$ %) ↑ |
|---|---|---|---|---|---|---|
| | | | MAE ↓ | RMSE ↓ | | |
| Dialogues | **S1 (LMM4-IC4K)** | **71.6±0.5** | $0.35 \pm 0.07$ | $2.81 \pm 0.08$ | **1.11±0.02** | **88.0±0.3** |
| | S2 | $63.5 \pm 0.4$ | **0.09±0.05** | **0.39±0.17** | $1.20 \pm 0.10$ | $82.5 \pm 0.1$ |
| | S3 | $62.6 \pm 0.4$ | $0.63 \pm 0.14$ | $5.60 \pm 0.04$ | $1.26 \pm 0.12$ | $85.6 \pm 0.1$ |
| | S4 | $31.3 \pm 0.5$ | $0.63 \pm 0.07$ | $5.67 \pm 0.04$ | $2.23 \pm 0.14$ | $75.6 \pm 0.1$ |
| | S5 | $25.4 \pm 0.3$ | $1.09 \pm 0.21$ | $10.27 \pm 0.06$ | $2.94 \pm 0.16$ | $74.6 \pm 0.2$ |
| Training Stages | T1 | $65.1 \pm 0.1$ | $0.59 \pm 0.06$ | $3.30 \pm 0.11$ | $1.17 \pm 0.03$ | $81.8 \pm 0.4$ |
| | T2 | $68.2 \pm 0.3$ | $0.41 \pm 0.05$ | $2.91 \pm 0.03$ | **1.03±0.06** | $85.3 \pm 0.3$ |
| | T3 | $24.7 \pm 0.5$ | $3.17 \pm 0.06$ | $36.99 \pm 0.33$ | $3.69 \pm 0.20$ | $64.9 \pm 0.2$ |
| | **T4 (LMM4-IC4K)** | **71.6±0.5** | **0.35±0.07** | **2.81±0.08** | $1.11 \pm 0.02$ | **88.0±0.3** |

over the industrial baseline underscores LMM4-IC4K's significant potential to advance the standardization of IC footprint labeling within the PCB industry.

To evaluate the gains that stem purely from domain-specific context, we further perform 4-shot experiments (one example per footprint layout type) on GPT-5, Gemini 2.5 Flash, and Qwen2-VL-7B, providing the models with geometric rules and datasheet examples of all variations of IC diagrams. As shown in Table 1, GPT-5's $IoU_{IC}$ increased from 40.6% to 43.0%, Gemini 2.5 Flash from 16.5% to 38.0%, and Qwen2-VL-7B from 1.7% to 3.8%, which still remains significantly lower than LMM4-IC4K's 71.6% $IoU_{IC}$. This indicates that while domain-specific context benefits this specialized reasoning task, specialized finetuning with a domain expert dataset is essential.

## 5.4. Comparison with Traditional CV Method

Traditional CV methods (e.g., object detection + OCR) are inadequate for IC footprint geometry labeling, as they focus on pattern recognition rather than the complex relational reasoning required between symbols and numeric annotations. Consequently, there have been no existing traditional CV-based solutions specifically for this task. To illustrate the limitations of conventional CV approaches, we developed a system combining object detection (Tian et al., 2025), OCR (Google, 2025), and rule-based methods aimed at recognizing pin geometries in IC diagrams (detailed in Appendix C.3). As shown in Table 1, this traditional system failed to interpret IC footprint geometries. The primary challenge lies in the lack of relational reasoning, where OCR-extracted numbers are not consistently placed near their corresponding geometric components, hindering correct associations. Additional rule-based pairing, omission handling, and complex calculations are required to generate accurate numeric descriptions. However, such rules are burdensome and incomplete under manual constraints. Our LMM approach leverages expert-labeled data to implicitly learn the rules necessary for accurate symbol-numeric annotation bounding and flexible geometry reasoning. By integrating relational reasoning, it overcomes the limita-

tions of traditional CV methods, providing more reliable and scalable solutions for IC footprint geometry labeling.

## 5.5. Ablation Study

### 5.5.1. DIFFERENT DIALOGUE TRAINING STRATEGIES

As described in Section 4, the problem of IC footprint geometry labeling can be divided into three distinct yet logically connected tasks: **pin number counting (task 1)**, **pin position understanding (task 2)**, and **pin dimension understanding (task 3)**. These tasks differ in complexity and interdependency. For instance, accurately counting pin numbers may be relatively straightforward, yet errors made at this initial stage will prevent correct solutions to the subsequent tasks. Considering these logical relationships, we evaluate the QA performance across various training strategies, reflecting different dialogue sequencing approaches.

We evaluate five training strategies: **S1**: training on all three tasks within a single conversation, with the tasks presented sequentially; **S2**: training on the three tasks sequentially, each within a separate conversation; **S3**: training on task 1 first, followed by tasks 2 and 3 within a single conversation; **S4**: training on tasks 1 and 2 within one conversation first, followed by tasks 1 and 3 within a separate conversation; **S5**: training for 3 rounds, each dedicated to a single task in the order of task 1, 2, 3. Please refer to Appendix B for a clearer description of the training strategies.

As shown in Table 2, the training strategy S1 achieves the highest $IoU_{IC}$ of 71.6%, increasing 12.8% compared with independent training in S2. This observation suggests that building a chain of thoughts among labeling tasks significantly enhances IC geometry understanding. The minor performance drop in Task 1 from S2 to S1 is negligible, as errors remain below one pin. This fluctuation is due to the simplicity of pin counting relative to complex downstream geometric tasks and does not significantly affect overall system performance. The sub-optimality of multi-round training leads to biased learning, where the model retains more recent task knowledge more effectively, while earlier knowledge gradually fades.

*Table 3.* Scaling analysis on real-world data volumes.

| Real Data Volume (%) | Overall ($IoU_{IC}$ %) ↑ | Task 1 (MAE) ↓ | Task 2 ($d_{pin}$ ↓) | Task 3 ($IoU_{pin}$ % ↑) |
|---|---|---|---|---|
| 0 (Syn Pre-Training) | 42.9 | 3.48 | 2.48 | 63.3 |
| 25 | 48.8 | 1.70 | 2.39 | 75.5 |
| 50 | 62.9 | 0.64 | 1.96 | 83.5 |
| 75 | 68.0 | 0.55 | 1.44 | 85.8 |
| **100 (LMM4-IC4K)** | **71.6** | **0.35** | **1.11** | **88.0** |

### 5.5.2. SYNTHETIC IMAGE DATASET BENEFITS

As stated in Section 4, LMM4-IC4K applies a two-stage training where it is trained under the synthetic part of IC-GEO8K in the first stage and the real-world part in the second stage. To explore the role of synthetic data in enhancing model performance, we evaluate the QA performance under various training strategies under different dataset part combinations as shown in Table 2: **T1**: training exclusively on real-world data; **T2**: training on a combined dataset of real-world and synthetic data simultaneously; **T3**: first training on real-world data (phase 1), then training on synthetic data (phase 2); **T4** (LMM4-IC4K): first training on synthetic data and subsequently fine-tuned on real-world data. T4 significantly improves performance, increasing $IoU_{IC}$ by 10%, 5%, and 189.9% compared to T1, T2, and T3, respectively. LMM4-IC4K gains a deeper understanding of real-world IC geometry by first learning general IC footprint patterns from synthetic data, and subsequently adapting to more complex real-world diagrams.

### 5.5.3. SCALING ANALYSIS ON REAL-WORLD DATA VOLUME

We further conduct a scaling analysis to evaluate the marginal utility of real-world data and the efficacy of synthetic pre-training. As shown in Table 3, pure synthetic data pre-training (0% real) already achieves an $IoU_{IC}$ of 42.9%, a substantial improvement over the 1.7% base model (Qwen2-VL-7B). This confirms that synthetic pre-training establishes a fundamental geometric prior that enables the model to resolve basic footprint patterns. The performance then scales with real-world data: 48.8% (25% real), 62.9% (50% real), and 68.0% (75% real), reaching 71.6% with the entire dataset. This progression demonstrates that while synthetic data provides a structural foundation, real-world samples are essential for calibrating the model to the stochastic noise, unit inconsistencies, and complex layout variations inherent in professional datasheets.

### 5.6. Comparison with Manual Labeling

To emphasize LMM4-IC4K's efficiency on IC footprint geometry labeling, we compare the effort of automated labeling using LMM4-IC4K with traditional manual labeling using EDA tools. We conduct labeling experiments on ICGEOQA with professional IC engineers using three common EDA software: Altium, Autodesk EAGLE, and Ki-

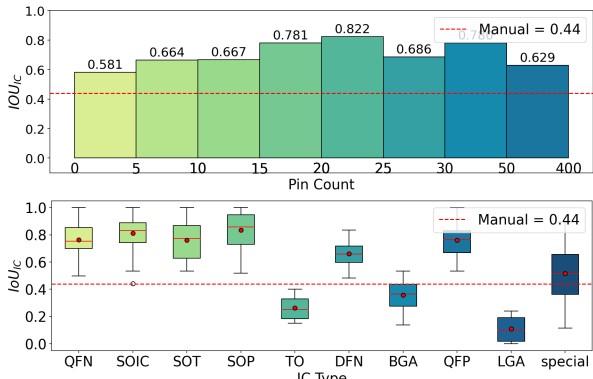

*Figure 6.* LMM4-IC4K performance across pin counts (upper) and IC types (lower).

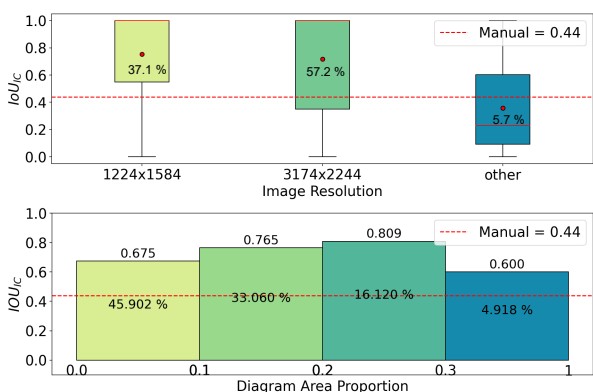

*Figure 7.* LMM4-IC4K performance across page image resolutions (upper) and IC diagram in-page proportions (lower). The percentages inside the boxes indicate the proportions of such data samples among all test samples.

CAD. The labeling times are recorded and averaged by the total sample number. We also record LMM4-IC4K's automated labeling time. Autodesk EAGLE and KiCAD require tens of minutes to manually process a single IC footprint diagram. Although Altium accelerates the manual labeling process by offering IC templates, it still takes an average of 7 $min/sample$. In contrast, LMM4-IC4K performs fully-automated labeling in just 15 $s/sample$ on 2 NVIDIA A100-40G GPUs, achieving a 28× speedup. Moreover, it reduces the per-sample cost from $0.61 (manual annotation) to $0.0014 (estimated via Qwen2-VL API (Aliyun, 2025)), yielding a 435× cost efficiency gain. These results demonstrate the substantial efficiency and scalability improvements enabled by LMM4-IC4K, marking a significant advancement in IC footprint labeling automation.

### 5.7. Robustness Analysis

We analyze **the performance of LMM4-IC4K on IC footprints across pin counts and IC types**. As shown in Figure 6 (upper), LMM4-IC4K achieves consistently high performance across various pin count ranges, demonstrating its robustness with respect to pin count variability.

Figure 6 (lower) shows that LMM4-IC4K achieves high performances across the majority of IC types. The under-performing types, namely TO, BGA, and LGA, are types that account for less than 10% of the whole IC modules (detailed in Appendix A). As these ICs are rare in the training samples, the model is less familiar with these types of footprints. This limitation can be reduced by collecting more samples on long-tail IC cases to improve the performance.

We further evaluate **the performance of LMM4-IC4K in relation to the image resolution and the proportion of IC diagrams within the page**. Since the images are sourced from real-world IC datasheets, they typically follow standard manuscript formats, ensuring consistent resolutions. As illustrated in Figure 7 (upper), only 5% of images deviate from common resolutions, indicating that the model performs well in the majority of cases. Most IC diagrams occupy less than 40% of the page area, with only 0.5% exceeding this threshold. As shown in Figure 7 (lower), LMM4-IC4K maintains high interpretation performance across various diagram proportions. While larger diagram layouts tend to lead to better performance, overly large diagrams often suggest more complex pin configurations, resulting in a performance decline for diagrams that occupy more than 30% of the page layout. However, such diagrams account for only 5% of the IC diagrams evaluated.

### 5.8. Failure Mode Analysis

We categorize failures into detectable geometric conflicts and logically consistent inaccuracies (examples can be found in Appendix C.6). For the former, such as pin orientation errors in dense layouts, the resulting inter-pin collisions provide a clear mathematical signal for post-processing filters. Additionally, shape-related errors typically occur when a dimension (*e.g.*, $dx$ or $dy$) is absent from the diagram, causing the model to default to a circular interpretation in which the only parameter is treated as the diameter. This can be mitigated by a dedicated shape classification module to enforce geometric priors. However, logically consistent inaccuracies, such as misassociating a numerical value with a nearby geometric parameter (*e.g.*, swapping pin gap $g$ with pin width $w$), are significantly harder to detect. Because the resulting footprint remains geometrically feasible and does not violate geometric constraints, these errors are often indeterminable without cross-referencing external labels or ground truth. This represents a fundamental challenge in technical drawing interpretation. While our current framework focuses on zero-meta-context geometric parsing, we acknowledge that incorporating external metadata (*e.g.*, IC names or package standards) to perform multi-modal consistency checks is a vital next step for industrial deployment. By establishing this high-precision baseline, our work provides the necessary foundation for such high-level semantic validation in future research.

## 6. Limitation and Discussion

**Cross-domain Diagram Understandings.** Unlike architectural or mechanical drawings, IC footprint diagrams follow strict industry standards (e.g., JEDEC (Standard, 2012), IPC-7351 (IPC, 2010)) and feature domain-specific elements like pin-1 markers and electrical connectivity constraints. These requirements demand millimeter-level precision to ensure manufacturing viability. Consequently, LMM4-IC4K is specifically optimized to address the unique geometric and structural complexities of IC footprints.

**Training Strategies.** Despite using only supervised fine-tuning on a 7B model (Qwen2-VL-7B), LMM4-IC4K outperforms frontier LMMs such as GPT-5 (by 76.4%) and Gemini-2.5 Flash (by 333.9%), validating our chain-of-thought design and expert VQA datasets. While Reinforcement Learning (*e.g.*(GRPO) (Shao et al., 2024)) or more complex agentic frameworks could further enhance results, we defer these to future work. Nonetheless, our research leverages the capabilities of general LMMs to interpret complex IC diagram geometry, advancing fully automated PCB engineering and contributing to the development of geometry-aware LMMs.

**Long-tail Failure Mitigation.** While our dataset reflects the high-frequency distribution of industrial IC libraries for immediate practical utility, extending robustness to long-tail package types (BGA, LGA, TO) is a vital next step. Preliminary balanced re-sampling and targeted synthetic augmentations on a BGA subset demonstrate that 3x synthetic augmentation combined with 2x balanced re-sampling has already improved $IoU_{IC}$ from 35% to 55%, confirming that balanced sampling and synthetic data can effectively provide the geometric priors missing in real-world distributions. These findings align with our scaling analysis (Section 5.5.3), which underscores the role of synthetic pre-training in establishing a structural foundation.

## 7. Conclusion

We present the first systematic investigation of general-purpose LMMs in understanding IC footprint geometry with the novel-proposed benchmark, ICGEOQA, revealing their limitations in precise spatial reasoning tasks. To address this, we develop LMM4-IC4K to enable LMMs to accurately interpret footprint diagrams and generate designs with SOTA performance. This methodology, supported by our expert-annotated ICGEO8K dataset, establishes a reproducible workflow for bridging the perceptual gap in IC footprint interpretation. By defining the expert reasoning chains necessary for this domain, we demonstrate how multimodal models can be adapted to high-precision, unstructured IC geometry where general-purpose approaches fail.

## Acknowledgements

We are grateful to all anonymous reviewers for their constructive comments. This work is supported in part by the United States National Science Foundation (NSF) (Grant No. CAREER-2046972), the National Natural Science Foundation of China (NSFC) (Grant No. 62502303), and the Natural Science Foundation of Shanghai (Grant No. 24ZR1439100, 25ZR1401171).

## Impact Statement

Our work leverages general LMMs to interpret complex IC diagram geometry, addressing a critical but overlooked gap in automated PCB design. By introducing an expert-annotated, pin-level dataset, we pave the way for standardized, fully automated PCB engineering and the advancement of geometry-aware LMMs. Additionally, our benchmark serves as a rigorous evaluation for LMMs in extracting complex numeric-symbolic relationships within technical domains.

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

# A. Data Collection

## A.1. Data distribution

Figure 8 shows the package type distributions of our datasets. Our proposed benchmark, ICGEOQA, matches the package type distribution of the 200K IC parts collected from Digi-Key (Electronics, 2025), demonstrating its representativeness of the real-world distribution within the IC community.

The real-world and synthetic subsets of our training dataset, ICGEO8K, share the same package type distribution, as they are derived from the same set of IC entries. Minor differences arise due to additional processing of real-world diagrams: diagrams with vague or irregular pins are excluded, reducing sample counts in certain categories. Moreover, some datasheet pages contain multiple target diagrams, which are split into separate samples, whereas each synthetic sample includes exactly one diagram generated from a single EDA description.

Figure 9 shows the pin count distributions of ICGEO8K and ICGEOQA. The real-world and synthetic subsets of ICGEO8K and ICGEOQA share the same pin count distribution.

## A.2. Benchmark data distribution

We downloaded 200K IC information entries, of which 50K have available datasheets. **IC EDA files:** Ultra Librarian (Ultra Librarian, 2024) is a comprehensive electronic component library offering access to 16 million verified components described in EDA design files. These files provide real-world examples of IC landing patterns, allowing extraction of numerical descriptions of IC parts (*i.e.*, pin positions and sizes). We download EDA design files corresponding to the collected IC datasheets. However, since EDA files are individually provided by manufacturers or engineers, over half of the datasheets in the Digi-Key library lack matching EDA files, reducing the pool of valid image-label pairs to fewer than 25K entries.

Figure 10 shows a variety of IC layout patterns for different package types. Different IC package types are largely distinct from each other in pin types, pin numbers, pin shapes, and placements, indicating the complexity of the IC geometry understanding problem.

Figure 5 illustrates the package type distribution of IC-GEOQA alongside the 200K IC footprint entries collected from Digi-Key. The distribution of ICGEOQA is deliberately designed to mirror that of the 200K IC footprint dataset, ensuring that our benchmark accurately reflects the practical requirements of PCB engineers in their daily design tasks.

Figure 9 (c) shows the distribution of pin numbers of our benchmark, indicating the diversity in pin count cases. As

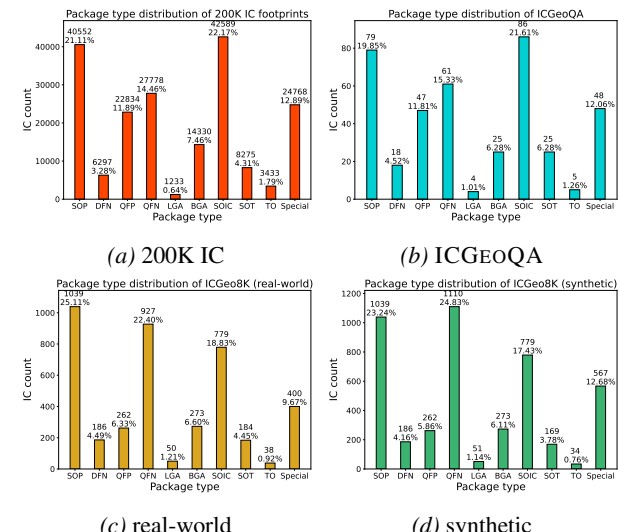

*(a)* 200K IC      *(b)* ICGEOQA

*(c)* real-world      *(d)* synthetic

*Figure 8.* IC package type distributions of datasets

over 80% of the samples contain less than 100 pins, it's reasonable to set the cut-off length of our model to 4096 to avoid massive token truncation.

## A.3. Data sample formation example

This section shows an example of the data processing pipeline for a dataset sample. Figure 11 shows the real-world datasheet pages of the target IC entry. In this example, the information about the IC footprint diagram is distributed over two pages. The two pages are concatenated into one image. The target diagram is located inside the red frame, while the associated parameter labels about the coordinates and dimensions of pins are framed in green and the corresponding values are framed in blue.

Figure 12 shows the EDA description of pins for the same IC entry. The EDA descriptions are in XML format and contain the location and dimension information of each pin. To visualize clearly, the pin indices are colored in blue, the coordinates of pins are colored in orange, and the dimensions of pins are colored in red.

Figure 13 shows the final data sample of the IC entry. The data sample comprises the datasheet page image and the JSON format conversation QAs. Note how the values in the three QAs are matched with those in Figure 12.

## A.4. Data Examples

This subsection presents example data samples from IC-GEO8K and ICGEOQA. Note that these samples represent raw data used for storage purposes and are not directly used for training or testing. The order of QAs shown here does not reflect the actual QA sequences used during training. The final training and testing samples are organized accord-

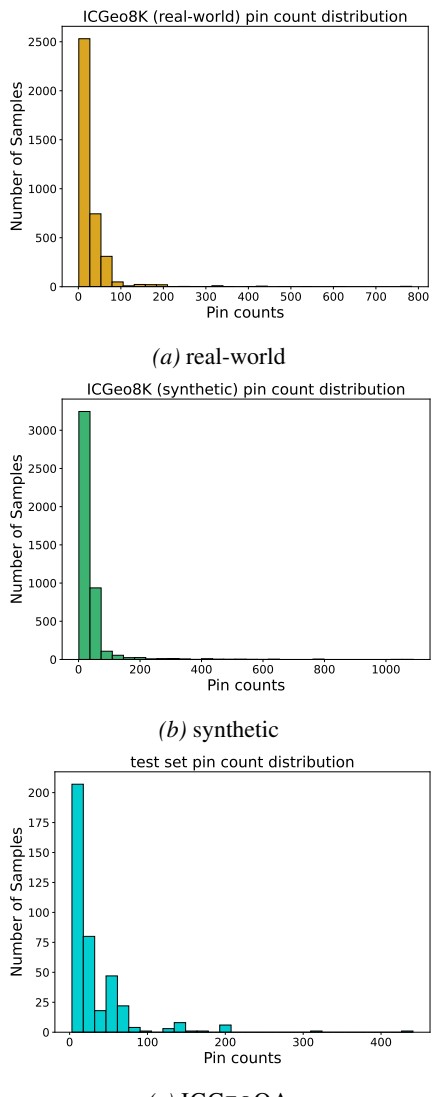

*(a)* real-world

*(b)* synthetic

*(c)* ICGEOQA

*Figure 9.* IC pin count distributions of datasets

ing to specific QA templates, as detailed in QA Examples.

A data sample in the dataset represents an image-text pair, where the image is either a datasheet page containing a suggested pad layout diagram or a synthesized diagram, and the text label consists of processed geometry descriptions in the form of three QAs: the number of pins, the pin coordinates, and the pin dimensions, as shown in Figure 14 and Figure 15. The datasets are organized in **sharegpt** format.

### A.5. Diagram Synthesis Toolkit

As stated in Synthetic Diagram Augmentation, we develop a footprint diagram generation toolkit that synthesizes clean, datasheet-style footprint images from imprecise EDA geometry descriptions.

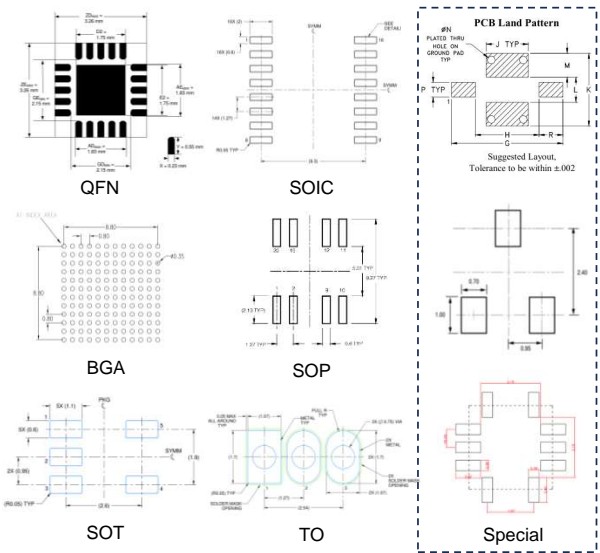

*Figure 10.* Examples of different diagram patterns for distinct package types.

The synthesis toolkit first reads the EDA descriptions of the IC entry and visualizes the IC layout based on the extracted data. To emulate the appearance of real suggested pad layout diagrams, it adds numeric annotations and auxiliary lines to replicate typical styles and formatting conventions. A comparison of the labeling styles and formats between synthetic and real-world diagrams is shown in Figure 16.

The count difference between the synthetic and the real-world samples arises from manual filtering in real-world samples, where real-world diagrams are removed if their pins are vague or irregular.

## B. QA Examples

As stated in Different Dialogue Training Strategies (Section 5.5.1), we evaluate the QA performance across various training strategies, reflecting different dialogue sequencing approaches. Each training strategy corresponds to a specific QA organization order, as detailed below:

- **Strategy 1 (S1)**: The QAs for the three subtasks are organized in a chain-of-thought manner: first querying the number of pins, followed by the coordinates of the pins, and finally the dimensions of the pins.

- **Strategy 2 (S2)**: The QAs for the three subtasks are separated into three individual data samples.

- **Strategy 3 (S3)**: The QAs for the three subtasks are divided into two training rounds: the first round includes samples that query only the number of pins, while the second round consists of samples that sequentially query the coordinates and dimensions of the pins.

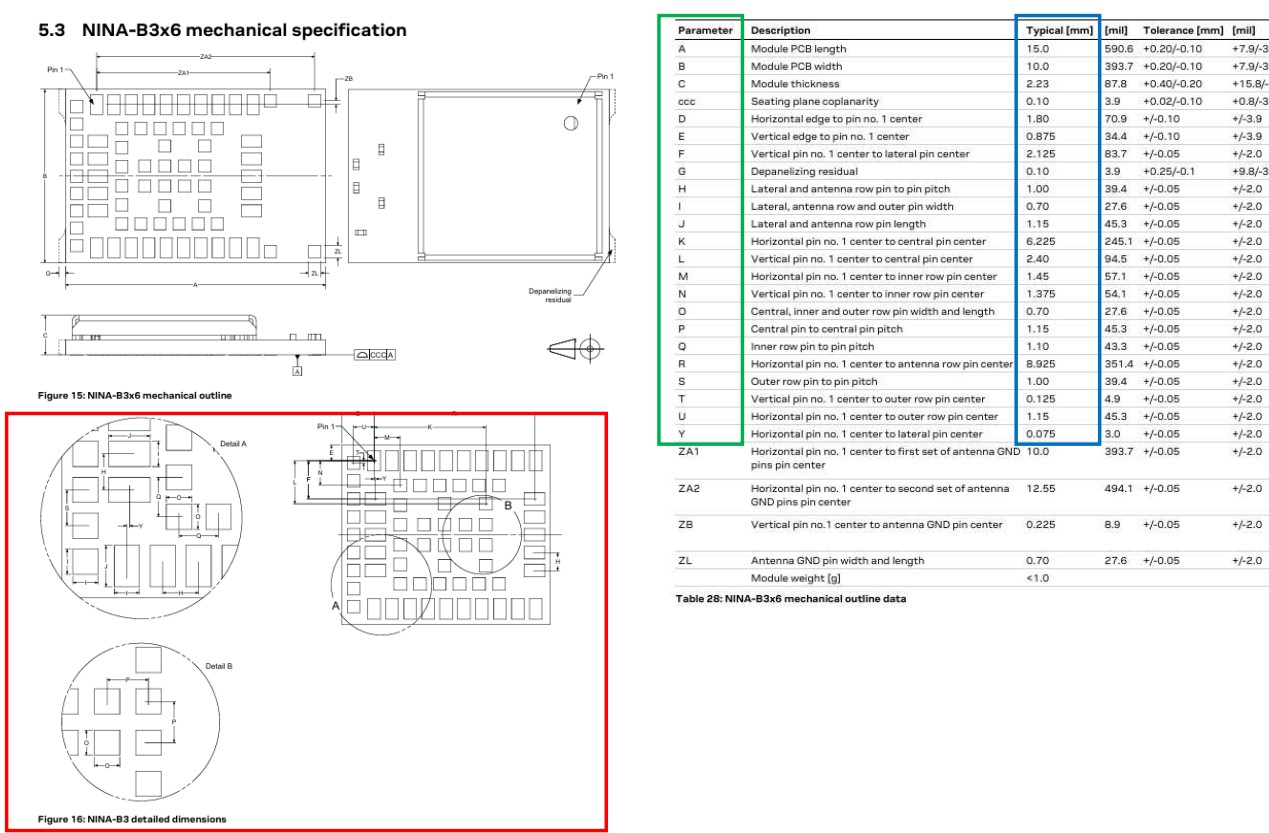

*Figure 11.* Real-world datasheet example.

*Figure 12.* EDA description example.

*Figure 13.* Dataset sample example.

- **Strategy 4 (S4)**: The QAs for the three subtasks are divided into two training rounds: the first round includes samples that sequentially query the number of pins and the coordinates of pins, while the second round consists of samples that sequentially query the number and dimensions of the pins.

- **Strategy 5 (S5)**: The QAs for the three subtasks are divided into three training rounds: the first round includes samples that query only the number of pins, the second round comprises samples that query only the pin coordinates, and the third round includes samples

that query only the pin dimensions.

The QA templates used to generate training and testing samples for each training strategy are detailed in Figure 17-Figure 21.

Some real-world datasheet pages contain more than one suggested pad layout diagram. For such cases, the data sample is separated into multiple data samples, each containing the identical datasheet image. The target diagram is specified in the prompt templates, as shown in Figure 22.

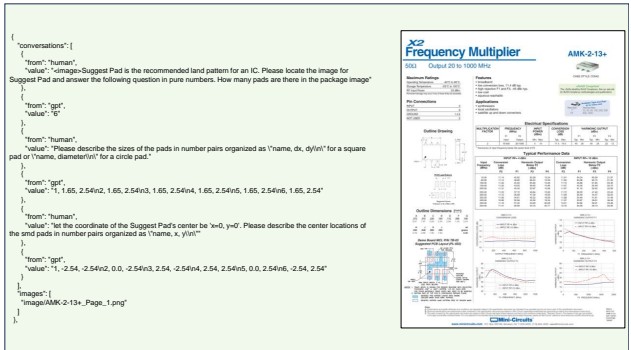

*Figure 14.* Data sample example 1. (Real-world Data)

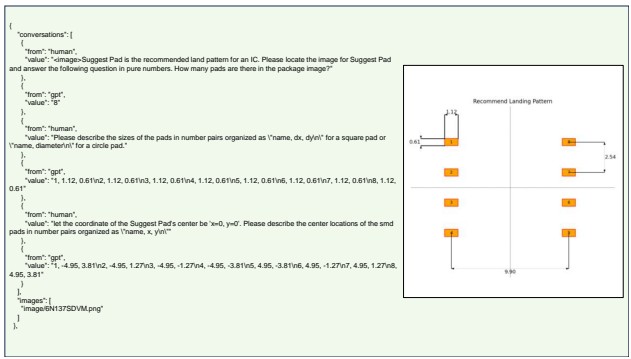

*Figure 15.* Data sample example 2. (Synthetic Data)

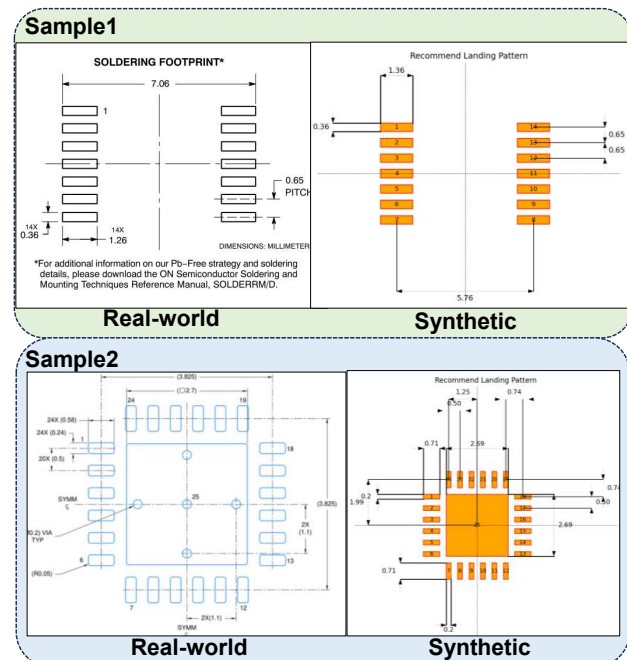

*Figure 16.* Examples of synthetic and real-world diagrams. Note that all key parameters to determine a pin layout are labeled in the synthetic diagram, and the pin indices are explicitly labeled in the pins' centers.

## C. Training Details and Additional Evaluation Details

### C.1. Training Details

The detailed training hyper-parameter settings of LMM4-IC4K during the two-stage fine-tuning process are shown in Table 4.

*Table 4.* Hyper-parameter settings for both training stages.

| Parameter | Value |
|---|---|
| Batch Size (per GPU) | 2 |
| Learning Rate (LR) | $5e^{-5}$ |
| LR Scheduler | cosine |
| Epoch | 3 |
| Cut-off Length | 4096 |
| Gradient Accumulation | 4 |
| Validation Set Ratio | 0.1 |
| Optimizer | AdamW |
| LoRA Rank | 8 |
| LoRA Alpha | 16 |
| LoRA Dropout | 0 |

### C.2. Manual Baseline

Our manual baseline is calculated using EDA descriptions from production-ready CAD models from UltraLibrarian (Ultra Librarian, 2024) and SnapEDA (SnapMagic Search, 2025), generated by the professional engineer community. Since the EDA description files corresponding to the benchmark ICs are directly sourced from industrial PCB libraries, they are considered sufficient to meet industry standards. However, as discussed in Section 3, engineers do not always adhere strictly to the specifications presented in footprint diagrams, resulting in discrepancies between datasheet diagrams and actual EDA annotations. While these offsets are common in practice, excessive mismatches cause assembly failures (e.g., desoldering). Consequently, the accuracy of aligning manual EDA descriptions with footprint diagram annotations serves as a practical baseline for evaluating industrial-standard functional labeling and usability criteria rather than a claim of surpassing human cognitive performance in diagram translation.

### C.3. Traditional CV Method Implementation

As stated in Section 5.4, we developed a traditional CV system combining object detection, OCR, and rule-based methods aimed at recognizing pin geometries in IC diagrams.

We begin by training an object detection model, YOLOv12 (Tian et al., 2025), to identify the region containing the suggested pad (IC footprint) diagram within the input image. After cropping the detected region, we apply morphological

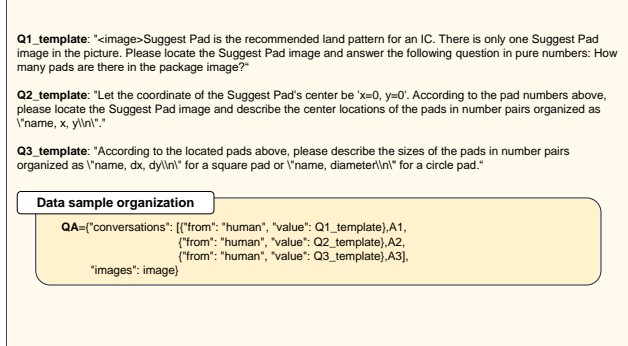

*Figure 17.* QA template for training strategy 1 (S1).

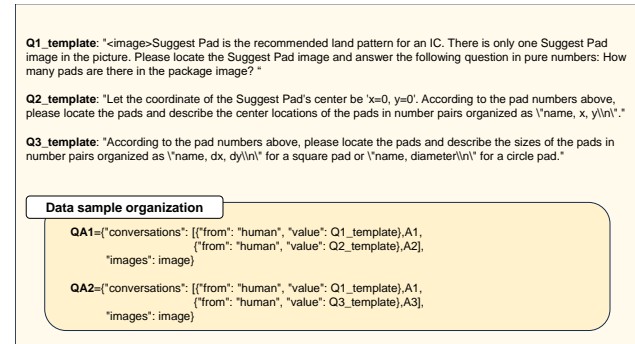

*Figure 19.* QA template for training strategy 3 (S3).

*Figure 18.* QA template for training strategy 2 (S2).

*Figure 20.* QA template for training strategy 4 (S4).

processing to extract the bounding box of the footprint. This bounding box enables us to determine the geometric center of the diagram, which is subsequently used as the coordinate origin for pin localization.

Next, we detect individual pins within the extracted diagram region by exploiting their geometric characteristics. Rectangular pins are identified as elongated, high-aspect-ratio boxes, whereas circular pins are recognized as round or near-round contours. To enhance robustness across different package types, we apply adaptive filtering that considers contour area, aspect ratio, rectangularity, and circularity, effectively suppressing auxiliary drawing elements (e.g., reference boxes, dashed guidelines) and random noise points. After this filtering stage, the remaining contours are classified as valid pins and subsequently used for estimating both pin dimensions and their coordinates relative to the diagram's center.

We employ Python–Tesseract OCR (madmaze, 2024) to extract candidate numeric annotations from the diagram. However, IC footprint diagrams typically contain numerous textual and dimensional labels, many of which are not positioned in close proximity to the corresponding pin graphics. As a result, directly associating OCR-extracted numbers with specific pin shapes is unreliable without additional semantic reasoning. Instead, we adopt an alternative strategy:

we identify at least one representative dimension annotation (*e.g.*, a labeled pitch or pad size) and measure its corresponding pixel length in the diagram. From this, we derive a pixel-to-millimeter scale factor. This scale factor is then applied uniformly across the entire diagram to convert all detected pin positions and dimensions from pixel units to millimeters.

Although the proposed traditional CV pipeline can extract pin-like contours from the diagram, it cannot reliably distinguish true pins from visually similar elements such as via holes, auxiliary rectangular markers, and decorative or noisy symbols. The geometric characteristics of pins and the corresponding filtering thresholds vary significantly across different images and package types, leading to inconsistent detection performance. In addition, footprint diagrams are schematic rather than metrically accurate representations of IC packages; thus, the spatial relationships inferred directly from the drawing often deviate from the numerical annotations provided in the diagram. Consequently, a purely contour-based approach is insufficient. A robust interpretation of IC footprint diagrams ultimately requires associating numeric annotations with symbolic diagram components and performing higher-level geometric reasoning to recover accurate pin dimensions and positions.

**Q1_template**: "<image>Suggest Pad is the recommended land pattern for an IC. There is only one Suggest Pad image in the picture. Please locate the Suggest Pad image and answer the following question in pure numbers: How many pads are there in the package image? "

**Q2_template**: "<image>Suggest Pad is the recommended land pattern for an IC. There is only one Suggest Pad image in the picture. Let the coordinate of the Suggest Pad's center be 'x=0, y=0'. Please locate the Suggest Pad image and describe the center locations of the pads in number pairs organized as \"name, x, y\\n\"."

**Q3_template**: "<image>Suggest Pad is the recommended land pattern for an IC. There is only one Suggest Pad image in the picture. Please locate the Suggest Pad image and describe the sizes of the pads in number pairs organized as \"name, dx, dy\\n\" for a square pad or \"name, diameter\\n\" for a circle pad."

**Data sample organization**

QA1={"conversations": [{"from": "human", "value": Q1_template},A1], "images": image}

QA2={"conversations": [{"from": "human", "value": Q2_template},A2], "images": image}

QA3={"conversations": [{"from": "human", "value": Q3_template},A3], "images": image}

Note: QA1, QA2, QA3 compose **three independent** datasets

*Figure 21.* QA template for training strategy 5 (S5).

**Q1_template_multi**: "<image>Suggest Pad is the recommended land pattern for an IC. There are {count} Suggest Pad images in the picture. Please locate the {place} Suggest Pad image and answer the following question in pure numbers: How many pads are there in the package image?"

**Q2_template_multi**: "Let the coordinate of the Suggest Pad's center be 'x=0, y=0'. According to the pad numbers above, please locate the {place} Suggest Pad image and describe the center locations of the pads in number pairs organized as \"name, x, y\\n\"."

**Q3_template_multi**: "According to the located pads above, please describe the sizes of the pads in number pairs organized as \"name, dx, dy\\n\" for a square pad or \"name, diameter\\n\" for a circle pad."

places=["first", "second", "third", "fourth"]

**Data sample organization**

QA={"conversations":
    [{"from": "human",
     "value":Q1_template_multi.format(count=diagram_count,place=diagram_place)},A1,
     {"from": "human",
     "value": Q2_template_multi.format(place=diagram_place)},A2,
     {"from": "human",
     "value": Q3_template_multi},A3],
     "images": image}

*Figure 22.* QA template for training strategy 1 (S1) with multiple diagrams. Please note the differences in the prompts compared with Figure 17.

## C.4. Comparison Details with Existing EDA tools

To emphasize LMM4-IC4K's efficiency on IC footprint geometry labeling, we compare the effort of automated labeling using LMM4-IC4K with traditional manual labeling using EDA tools. We conduct labeling experiments on ICGEOQA with 11 professional IC engineers using three common EDA software: Altium (Team, 2022), Autodesk EAGLE (Autodesk, Inc., 2020), and KiCAD (SparkFun Electronics, n.d.). These tools are widely used in the electronics industry for schematic design and PCB layout, each offering distinct workflows and footprint generation capabilities.

Altium Designer is a professional-grade EDA tool known for its comprehensive feature set, advanced routing tools, and tight integration of schematic and layout views, often used in high-end commercial and industrial applications. It provides various footprint templates to enable fast and accurate footprint drawing. Autodesk EAGLE is a popular lightweight tool favored by startups and hobbyists for its intuitive interface and strong integration with Autodesk Fusion 360 for mechanical-electronic co-design. KiCAD, an open-source EDA suite, is widely adopted in academia and open hardware communities for its flexibility, active community support, and full-featured PCB layout and schematic

editing capabilities without licensing fees.

In this user study, each engineer is tasked with manually labeling a set of IC footprints using their preferred EDA tool, while we record the time taken for each labeling task to compare performance between LMM4-IC4K and existing EDA tools. Each engineer is provided with the identical predefined list of IC footprint samples from the ICGEOQA benchmark and is asked to manually implement the corresponding footprints using their preferred EDA tool.

## C.5. Qualitative Evaluation

In this subsection, we present comprehensive IC footprint geometry cases to demonstrate the capability of LMM4-IC4K in understanding IC diagram geometries. We compare the reasoning results of LMM4-IC4K with those of SOTA general-purpose LMMs, including GPT-4o (Hurst et al., 2024), Gemini 2.0 (Mallick & Kilpatrick, 2025), and DeepSeek-VL2 (Wu et al., 2024). To better illustrate the models' outputs, the reasoning results are visualized and compared against the ground-truth pin layouts, as shown in Figure 23.

As illustrated in Figure 23, LMM4-IC4K demonstrates accurate understanding across diverse diagram cases featuring varying land patterns, pin counts, pin shapes, and datasheet image conditions. LMM4-IC4K consistently predicts the correct number of pins, their coordinates, and dimensions, resulting in perfect alignment with the ground-truth pin layouts. In Figure 23(b), LMM4-IC4K accurately identifies the coordinates and diameters of the round pins, whereas GPT-4o and Gemini 2.0 produce incorrect pin dimensions, and DeepSeek-VL2 mistakes the round pins as rectangular. In Figure 23(d), all three general-purpose LMMs fail to predict the correct pin coordinates, while LMM4-IC4K correctly recognizes the target diagram in the datasheet image and outputs precise locations for all pins. These case studies further highlight the capability of our model in understanding IC footprint geometries.

## C.6. Failure Analysis

In this subsection, we present some typical failure cases of LMM4-IC4K, as shown in Figure 24.

In one typical case (Figure 24 (a) ), the widths and heights of rectangular pins are reversely interpreted. This may occur to the x and y coordinates as well. This failure mode is typical in 4-side pin arrangements, where the pins have the same dimensions but with different rotations. This failure mode may be resolved with collision detection among pins.

Another typical failure case is illustrated in Figure 24 (b), where circular pins are mistaken for square pins. This failure mode may be a result of an imbalance in the training samples of circular and rectangular pins. Irregular pin arrangements

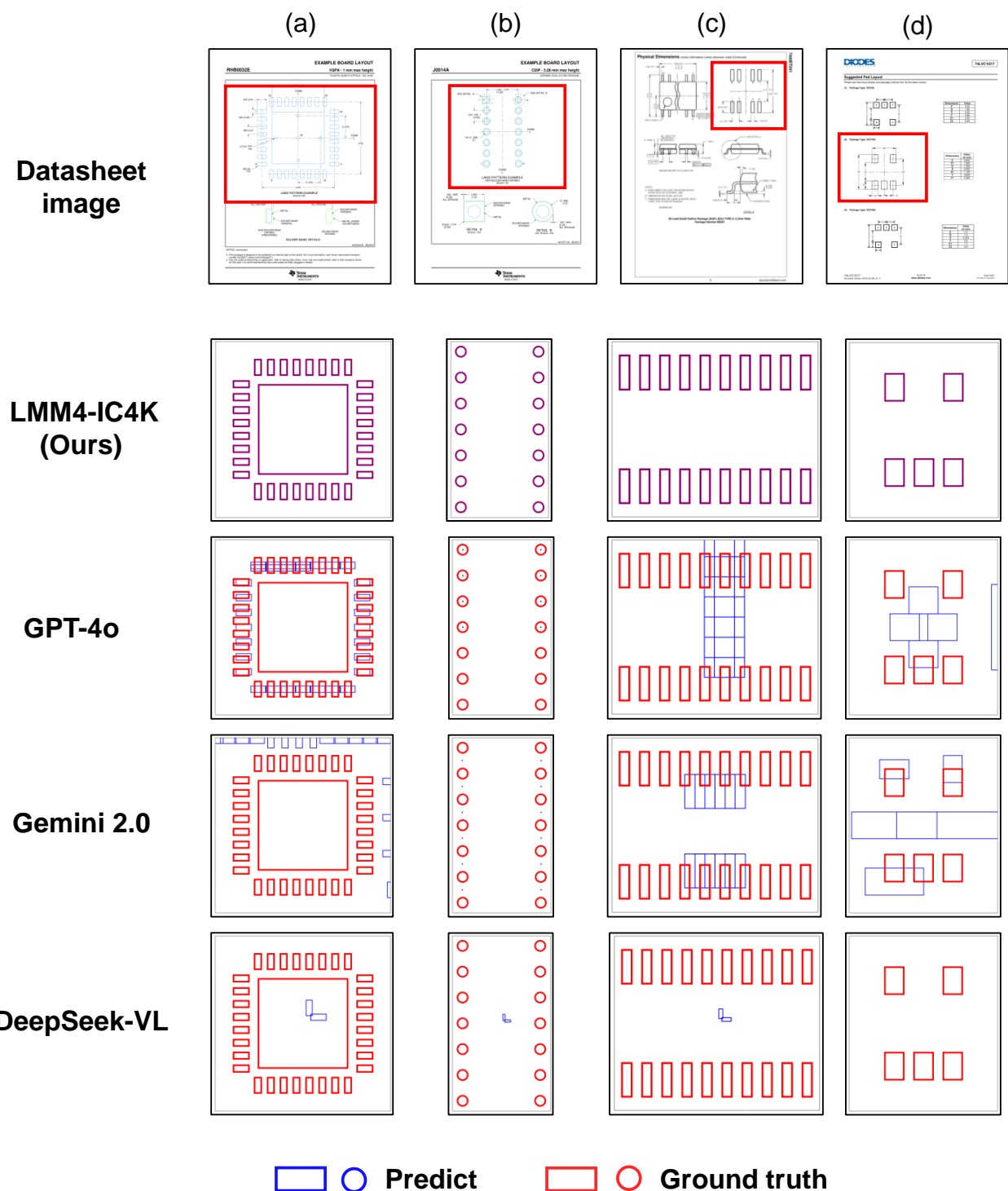

*Figure 23.* IC footprint geometry understanding examples with different methods. The target diagrams in the datasheet pages are highlighted in red frames.

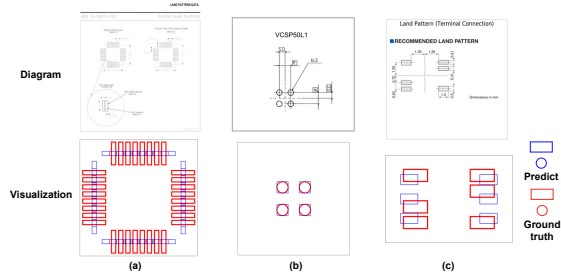

*Figure 24.* Typical failure cases of LMM4-IC4K. (a) An example where the widths and heights of pins are reversed. (b) An example of circular pins mistaken for square pins. (c) An example of an irregular pin arrangement mistaken as a symmetrical arrangement.

(exemplified in Figure 24 (c)) are hard to interpret as they are lacking in training samples and are asymmetrical, hence require extra reasoning awareness. Further expansion of the training dataset may alleviate these failure modes.

### C.7. Computational Complexity Analysis

The number of visual tokens corresponds to the input image size, which is relatively consistent owing to the consistent image resolution of datasheet PDFs.

The training time of LMM4-IC4K is approximately 2 hours for each training stage. The inference time of LMM4-IC4K is approximately 4.5 $s/sample$. Adding up the image up-loading time and description generation time, the total label-ing time is 15 $s/sample$. All computational analyses are performed on 2 NVIDIA A100-40G GPUs and 24 Intel(R) Xeon(R) Gold 6248R CPU @ 3.00GHz.

