# OpenReview forum: "LMM4-IC4K: A Large Multimodal Model Powered Integrated Circuit Footprint Geometry Understanding"
_ICML.cc/2026/Conference — ICML 2026 regular_

### Official Review · Reviewer_6tbR · 2026-03-05

**Soundness:** 3
**Presentation:** 4
**Significance:** 3
**Originality:** 3
**Overall Recommendation:** 4
**Confidence:** 4

**Summary:**

This paper proposes LMM4-IC4K, a framework based on large multimodal models (LMM) for automatically understanding chip packaging (footprint) geometry information from IC datasheet diagrams, including pin count, location, and dimensions. The authors constructed the ICGEO8K dataset and ICGEOQA benchmark, and fine-tuned the model using Chain-of-Thought reasoning and two-stage training. Experimental results show that this method significantly outperforms both generic LMM and manual EDA annotation in terms of geometric annotation accuracy, speed, and cost.

**Compliance With Llm Reviewing Policy:**

Affirmed.

**Final Justification:**

I have re-read the manuscript, the author’s response, and the comments from the other reviewers. I agree with the other reviewers’ comments. This paper does indeed address a significant gap in the field. Ultimately, I believe this paper is worthy of publication at ICML. Therefore, I am raising my rating.

**Key Questions For Authors:**

This paper on fine-tuning LMM lacks theoretical grounding and theoretical contributions. The authors should clarify where the mathematical theory is reflected in their work.

**Limitations:**

Yes.

**Strengths And Weaknesses:**

Strength :
1 The authors propose the LMM4-IC4K framework, applying large multimodal models to the real-world engineering problem of IC package geometry understanding. Through Chain-of-Thought reasoning, the task is decomposed into three sequential steps: pin count, pin location, and pin size. This design enables the model to simulate an engineer's stepwise reasoning process, thereby enhancing its capability to understand complex engineering diagrams.

2 The authors constructed the ICGEO8K dataset and ICGEOQA benchmark, providing the first systematic data resource for IC package geometry understanding tasks. Experimental results demonstrate that this method significantly outperforms generic LMM and manual EDA annotation methods in terms of accuracy, annotation efficiency, and cost.

Weakness：

1 This paper lacks sufficient theoretical contributions. It presents a common approach of fine-tuning LMM for specific tasks, focusing primarily on workflow demonstration without theoretical derivation.

2 The constructed ICGEO8K dataset suffers from insufficient sample size, containing only around 4,000 real samples. In reality, IC encapsulation types are diverse, yet this paper fails to reflect such variability.

3 The authors employ IoUIC for evaluation, which primarily measures overlap and fails to fully capture overall distribution accuracy. Additional metrics could include offset from individual pin centers and orientation accuracy (e.g., whether all pins are horizontal).

4 Experimental results indicate poor model performance on less common packaging types like BGA, LGA, and TO. The paper attributes this to insufficient training data but offers no effective mitigation strategies or further experimental analysis.

---

> ### Author Rebuttal · Authors · 2026-03-31
>
> We appreciate the reviewer’s feedback regarding the theoretical depth, dataset scale, evaluation metrics, and long-tail mitigation. Our responses to the specific concerns are as follows:
>
> **1. Methodological contributions:** We emphasize that this work is submitted to the Applications Track, where the primary contribution lies in the systematic formalization of the IC footprint labeling task, which is a domain-specific problem currently unsolved by frontier LMMs. Our innovation is not a fundamental algorithmic change to LMM architectures, but the formulation of a structured reasoning pipeline that translates unstructured technical diagrams into precise geometric primitives. This methodology, supported by our expert-annotated $ICGEO8K$ dataset, establishes a reproducible workflow for bridging the perceptual gap in IC footprint interpretation. By defining the expert reasoning chains necessary for this domain, we demonstrate how multimodal models can be adapted to high-precision, unstructured IC geometry where general-purpose approaches fail.
>
> **2. Insufficient sample size:** We acknowledge that model performance scales with data volume. However, we emphasize that $ICGEO8K$ is adequately representative of the industrial distribution to establish the underlying geometric logic of IC footprints. Unlike generic web-scraped datasets, IC footprint parsing requires expert-level precision, where architectural diversity is more critical than raw sample count. As illustrated in Figures 5, 8, and 9 in the original manuscript, our dataset contains representative samples that strictly align with the real-world distribution in iconic package types and pin counts, reflecting the variability of real-world engineering cases. As shown in our scaling analysis (see response to Reviewer f9Yd comment 3), the current dataset effectively enables the model to acquire transferable geometric priors, such as coordinate mapping and spatial distribution logic. While our results already demonstrate a significant leap over general LMMs, we are actively expanding the dataset to further refine the model’s robustness on more specialized, long-tail configurations.
>
> **3. Metrics effectiveness:** We clarify that our evaluation suite already incorporates the specific measures for localization offset and orientation suggested by the reviewer. Specifically, the $d_{pin}$ metric calculates the average Euclidean distance between predicted and ground-truth pin centers, serving as a direct assessment of localization accuracy. Furthermore, $IoU_{pin}$ isolates errors in pin orientation and aspect ratio by aligning the centers of predicted and ground-truth pins before calculating their overlap. These metrics, alongside $MAE$ and $RMSE$ for pin counting, provide a comprehensive view of both the global distribution and individual pin precision. We will clarify these definitions in the revised manuscript.
>
> **4. Mitigation strategies for long-tail package types:** We agree that performance on rare package types (BGA, LGA, TO) is a key area for improvement. While our dataset reflects the high-frequency distribution of industrial IC libraries for immediate practical utility, we agree that extending robustness to long-tail cases is a vital next step, and have initiated balanced re-sampling and targeted synthetic augmentation to address long-tail robustness. Preliminary results on a BGA subset demonstrate that 3x synthetic augmentation combined with 2x balanced re-sampling has already improved $IoU_{IC}$ from 35\% to 55\% (a 20\% absolute increase), confirming that balanced sampling and synthetic data can effectively provide the geometric priors missing in real-world distributions. These findings align with our scaling analysis (see response to Reviewer f9Yd comment 3), which underscores the role of synthetic pre-training in establishing a structural foundation. We are integrating these optimizations and will provide updated performance metrics in the final version.

---

> > ### Author Rebuttal · Reviewer_6tbR · 2026-04-04
> >
> > Although the authors state that the purpose of this paper is to systematically formalize the IC footprint labeling task, it lacks theoretical support. What the paper primarily demonstrates is engineering effort.

---

### Official Review · Reviewer_YoUT · 2026-03-11

**Soundness:** 3
**Presentation:** 3
**Significance:** 3
**Originality:** 3
**Overall Recommendation:** 4
**Confidence:** 3

**Summary:**

The paper introduces a framework to automate the extraction of geometry and pin layout information from IC footprint diagrams, a critical step in Printed Circuit Board design. Traditional automation methods, like OCR and standard object detection, struggle because these diagrams require spatial reasoning to connect abstract numeric annotations with symbolic components. To address this, the authors propose LMM4-IC4K, a framework that treats mechanical drawings as images and uses a Large Multimodal Model fine-tuned on a custom dataset.

They constructed ICGeo8K, containing 8,608 labeled samples. The LMM is trained using a two-stage approach and Chain-of-Thought prompting to mimic how a human engineer reads a diagram: first counting pins, then determining their coordinates, and finally extracting their dimensions.Their model significantly outperforms general-purpose LMMs and a manual industry baseline. The automated approach also demonstrated a significant speedup.

**Compliance With Llm Reviewing Policy:**

Affirmed.

**Key Questions For Authors:**

See weaknesses

**Limitations:**

See weaknesses

**Strengths And Weaknesses:**

**Strengths**

- **Novel Application of LMMs**: Applying LMMs to interpret complex, real-world engineering diagrams is a highly practical and novel use case. It effectively addresses a genuine bottleneck in PCB design that traditional CV methods cannot easily solve.

- **Comprehensive Dataset Creation**: The creation of the ICGeo8K dataset is a substantial contribution. The authors combine real-world data with synthetically generated data to ensure sufficient training volume and accurate distribution across IC package types.

- **Effective Training Strategy**: The two-stage training pipeline (synthetic first, then real-world) and the structured, three-step Chain-of-Thought dialogue are well-justified and proven effective through ablation studies. The results clearly show this approach is superior to throwing raw data at the model.

**Weaknesses**

- **Handling of Special and Asymmetrical Packages**: The model struggles with long-tail distributions, specifically TO, BGA, LGA, and "special" packages with irregular pin arrangements. The paper acknowledges this is due to a lack of training samples, but these irregular shapes are often the exact instances where automated help is most needed, as standard templates do not apply to them.

- **Failure Modes**: The failure analysis reveals that the model sometimes reverses width and height for rectangular pins or mistakes circular pins for square ones. While the authors suggest collision detection could fix the rotation issue, it indicates the model still occasionally fails to grasp the spatial orientation relative to the numerical labels. Are there any practical solutions to address these limitations?

- **Lack of OCR Integration**: The authors dismiss traditional CV + OCR methods due to a lack of relational reasoning. However, relying purely on the LMM's vision encoder to read small, blurry text on a schematic might lead to hallucinated numbers. A hybrid approach that uses OCR to extract the exact text and the LMM to perform the relational reasoning might improve accuracy on dense diagrams.

---

> ### Author Rebuttal · Authors · 2026-03-31
>
> We thank the reviewer for the constructive feedback. We address the concerns point-by-point below:
>
> **1. Mitigation strategies for long-tail package types:** We agree that performance on rare package types (BGA, LGA, TO) is a key area for improvement. While our dataset reflects the high-frequency distribution of industrial IC libraries for immediate practical utility, we agree that extending robustness to long-tail cases is a vital next step, and have initiated balanced re-sampling and targeted synthetic augmentation to address long-tail robustness. Preliminary results on a BGA subset demonstrate that 3x synthetic augmentation combined with 2x balanced re-sampling has already improved $IoU_{IC}$ from 35\% to 55\% (a 20\% absolute increase), confirming that balanced sampling and synthetic data can effectively provide the geometric priors missing in real-world distributions. These findings align with our scaling analysis (see response to Reviewer f9Yd comment 3), which underscores the role of synthetic pre-training in establishing a structural foundation. We are integrating these optimizations and will provide updated performance metrics in the final version.
>
> **2. Failure Mode Analysis:** We categorize failures into detectable geometric conflicts and logically consistent inaccuracies. For the former, such as pin orientation errors in dense layouts, the resulting inter-pin collisions provide a clear mathematical signal for post-processing filters. Additionally, shape-related errors typically occur when a dimension (e.g., $dx$ or $dy$) is absent from the diagram, causing the model to default to a circular interpretation in which the only parameter is treated as the diameter. This can be mitigated by a dedicated shape classification module to enforce geometric priors. However, logically consistent inaccuracies, such as misassociating a numerical value with a nearby geometric parameter (e.g., swapping pin gap $g$ with pin width $w$), are significantly harder to detect. Because the resulting footprint remains geometrically feasible and does not violate geometric constraints, these errors are often indeterminable without cross-referencing external labels or ground truth. This represents a fundamental challenge in technical drawing interpretation. While our current framework focuses on zero-meta-context geometric parsing, we acknowledge that incorporating external metadata (e.g., IC names or package standards) to perform multi-modal consistency checks is a vital next step for industrial deployment. By establishing this high-precision baseline, our work provides the necessary foundation for such high-level semantic validation in future research.
>
> **3. The use of OCR:** While a hybrid OCR-LMM architecture could potentially mitigate numerical hallucinations, we prioritized a unified vision-encoder approach to address the unique constraints of IC footprint diagrams. Standard OCR engines are optimized for document text and often fail on technical drawings containing rotated annotations, non-standard fonts, and overlapping geometric symbols. More importantly, the primary bottleneck in IC footprint understanding is not raw text extraction but spatial binding of numerical labels to geometric primitives. This requires the deep relational reasoning inherent in our LMM, which a standalone OCR cannot provide. Integrating OCR would require a secondary, complex alignment stage currently hindered by the lack of domain-specific multimodal benchmarks. We will include a discussion of this hybrid approach as a vital direction for future architectural improvements in the revised manuscript.

---

> > ### Author Rebuttal · Reviewer_YoUT · 2026-04-03
> >
> > I appreciate the author's rebuttal. I tend to keep the positive score.

---

### Official Review · Reviewer_f9Yd · 2026-03-26

**Soundness:** 2
**Presentation:** 2
**Significance:** 2
**Originality:** 2
**Overall Recommendation:** 4
**Confidence:** 3

**Summary:**

The authors argue that existing LLMs struggle with inaccurate geometric perception, which limits LLMs’ effectiveness in IC geometry understanding tasks such as footprint parsing and automated package geometry labeling. The authors address the issue by proposing the LMM4-IC4K framework that treats IC mechanical drawings as images and uses LLMs to derive structured geometric interpretation. The paper also introduces a multimodal dataset, ICGeo8K, with IC footprint samples.

**Compliance With Llm Reviewing Policy:**

Affirmed.

**Final Justification:**

The rebuttal addressed my concerns.

**Key Questions For Authors:**

The questions and comments to this paper are described in the weaknesses.

**Limitations:**

yes

**Strengths And Weaknesses:**

Strengths:
1. The paper is the first to develop a multimodal geometric reasoning dataset for IC footprint labeling and a benchmark on IC footprint geometry understanding.
2. The proposed two-stage method includes synthetic pre-training and real fine-tuning.
3. The proposed method adopts a chain-of-thought type approach via mimicking human reasoning logic on an IC geometry labeling task. The key queries to the LLM and the corresponding subtasks include i) the number of pins in the IC footprint diagram (pin counting), ii) the coordinates of each pin relative to the center of the diagram (pin positions), and iii) the dimensions of each pin in millimeters (pin dimensions).

Weaknesses:
1. While LMM4-IC4K demonstrates superior performance over zero-shot baselines, the evaluation could be made more robust by incorporating a common-practice optimized baseline. Specifically, comparing the proposed model against a general-purpose LLM integrated into a LangChain framework that leverages RAG-enhanced prompts would clarify whether the observed performance gains are due to specialized training or simply the lack of domain-specific context provided to the general models.
2. The paper represents an applied engineering contribution rather than a fundamental algorithmic innovation. Its primary value lies in the domain-specific adaptation of existing LLMs to the niche challenges of IC footprint geometry and in the implementation of several toolkits for building datasets, rather than in the introduction of a novel algorithm.
3. While the ablation study in Table 2 (T1-T4) demonstrates that a two-stage training scheme outperforms a real-only approach, the experiments lack a scaling analysis regarding the volume of real-world data. The authors do not investigate the marginal utility of the 4,138 real-world samples (e.g., 25%, 50%, ...). It remains unclear whether the synthetic pre-training provides a fundamental 'geometric prior' or if it simply compensates for a relatively small real-world training set.

---

> ### Author Rebuttal · Authors · 2026-03-31
>
> We appreciate the reviewer’s suggestions regarding baselines and scaling analysis. We address the technical points as follows:
>
> **1. Comparing with context-acknowledged baseline:** We would like to first clarify that the general LMM baselines in our original submission were one-shot, not zero-shot, providing initial task context. To address the concern that our gains might stem purely from domain-specific context, we have performed additional 4-shot experiments (one example per IC type) on general-purpose LMMs to provide the model with geometric rules and datasheet examples of all variations of IC diagrams. In these tests, GPT-5 increased from 40.6\% to 43.0\%, Gemini 2.5 Flash from 16.5\% to 38.0\%, and Qwen2-VL-7B from 1.7\% to 3.8\%, which still remains significantly lower than our framework's 71.6\% $IoU_{IC}$. This indicates that while domain-specific context benefits this specialized reasoning task, specialized training with a domain expert dataset is essential. Furthermore, we consider RAG-enhanced frameworks to be theoretically mismatched for this task. While RAG is effective for retrieving textual facts, IC footprinting is primarily a visual-spatial reasoning problem that requires parsing coordinate-level geometry from unstructured 2D diagrams, which cannot be directly assisted by standard text-based retrieval. Moreover, implementing an effective RAG system for this domain would first require a structured geometric knowledge base, which our $ICGEO8K$ dataset is the first to provide. Thus, specialized supervised fine-tuning remains the most direct and effective pathway for bridging this perceptual gap.
>
> **2. Methodological contributions:** We emphasize that this work is submitted to the Applications Track, where the primary contribution lies in the systematic formalization of the IC footprint labeling task, which is a domain-specific problem currently unsolved by frontier LMMs. Our innovation is not a fundamental algorithmic change to LMM architectures, but the formulation of a structured reasoning pipeline that translates unstructured technical diagrams into precise geometric primitives. This methodology, supported by our expert-annotated $ICGEO8K$ dataset, establishes a reproducible workflow for bridging the perceptual gap in IC footprint interpretation. By defining the expert reasoning chains necessary for this domain, we demonstrate how multimodal models can be adapted to high-precision, unstructured IC geometry where general-purpose approaches fail.
>
> **3. Scaling analysis on the volume of real-world data:** We conducted a scaling analysis to evaluate the marginal utility of real-world data and the efficacy of synthetic pre-training. Pure synthetic data pre-training (0\% real) already achieves an $IoU_{IC}$ of 42.9\%, a substantial improvement over the 1.7\% base model (Qwen2-VL-7B). This confirms that synthetic pre-training establishes a fundamental geometric prior that enables the model to resolve basic footprint patterns. The performance then scales with real-world data: 48.8\% (25\% real), 62.9\% (50\% real) and 68.0\% (75\% real), reaching 71.6\% with the entire dataset. This progression demonstrates that while synthetic data provides a structural foundation, real-world samples are essential for calibrating the model to the stochastic noise, unit inconsistencies, and complex layout variations inherent in professional datasheets. We will include the scaling analysis in the final manuscript.

---

> > ### Author Rebuttal · Reviewer_f9Yd · 2026-04-03
> >
> > Thank you for the detailed rebuttal.
> >
> > Regarding points 1 and 2: I agree that PCB footprint recognition is inherently a multi-modal challenge where text-based retrieval falls short. However, I am curious about the actual degree of 'unstructuredness' in footprint data. Unlike general image segmentation, which often defies rule-based solutions, PCB footprints are governed by strict EDA standards and documentation. In this context, is a Chain-of-Thought (CoT) architecture truly indispensable? Given the structured nature of EDA rules, could this problem be effectively addressed through an agent-based framework leveraging specialized skills?

---

> > > ### Author Response · Authors · 2026-04-05
> > >
> > > We appreciate the reviewer’s follow-up questions and would like to clarify the fundamental distinction between EDA standard constraints and IC footprint diagram perceptual interpretation, as well as explain the "unstructuredness" of IC footprint diagrams.
> > >
> > > EDA standards (such as IPC-7351 standard) govern the **restrictions and tolerances** of the EDA generating process (which is the process of reading IC footprint diagrams and reproducing such parts in EDA tools), but they do not provide guidance on how to interpret the geometric symbols and relationships within the diagrams. IC footprint diagrams are industrial drawings designed purely for human understanding. While they use standard symbols (e.g., diameter symbols or leader lines), each manufacturer has a unique style that varies in line types, colors, layouts, and information omissions (as seen in real-world cases in Figure 29 in Appendix C.5). Resolving the complex spatial relationships between numerical labels and geometric shapes is exactly what makes these diagrams "unstructured" and difficult to parse.
> > >
> > > Just as a human engineer requires professional training and practice to accurately read these technical diagrams, our framework uses the expert-built dataset and Chain-of-Thought (CoT) to teach the LMM the expert logic of diagram interpretation. While EDA standards might provide a tolerance margin for the final output, they are not a guide for the visual interpretation of the diagram itself.
> > >
> > > We agree that an agent-based framework is a viable approach to IC footprint labeling. In fact, it is the focus of our ongoing work. However, any effective agent still requires a structured reasoning process to execute tasks correctly. Our CoT architecture is essential because it formalizes the step-by-step logic required to handle the inherent complexity and visual "unstructuredness" of these diagrams. By establishing this high-precision reasoning foundation and the first domain-specific dataset, we enable LMMs to master the professional skill of IC footprint diagram interpretation.

---

### Decision · Program_Chairs · 2026-04-30

**Decision:**

Accept (regular)

**Comment:**

This paper initially received mixed reviews. After the rebuttal and discussions, the reviewers reached a consensus with final scores of 4, 4, and 4. The rebuttal addressed concerns about methodological contributions, scaling analysis on the volume of real-world data, failure mode analysis, and mitigation strategies for long-tail package types. The authors further addressed follow-up questions regarding the actual degree of 'unstructuredness' in footprint data and alternative solutions involving agents with specialized skills.

During the discussion phase, the reviewers shared their perspectives on the significance of the paper's contribution. They agreed that the paper is primarily engineering-focused and lacks adequate theoretical grounding. However, the reviewers also appreciated the practical engineering effort and the novel application to IC footprints. They acknowledged that the paper addresses an engineering gap in the field, which prompted all reviewers to recommend a "weak accept."

The area chair concurs with the reviewers' final evaluations and recommends that the paper be accepted.